# Urinary Metabolomics Study on the Protective Role of Cocoa in Zucker Diabetic Rats via ^1^H-NMR-Based Approach

**DOI:** 10.3390/nu14194127

**Published:** 2022-10-04

**Authors:** Elisa Fernández-Millán, Sonia Ramos, David Álvarez-Cilleros, Sara Samino, Nuria Amigó, Xavier Correig, Mónica Chagoyen, Carmen Álvarez, María Ángeles Martín

**Affiliations:** 1Department of Biochemistry and Molecular Biology, Faculty of Pharmacy (UCM), 28040 Madrid, Spain; 2CIBER of Diabetes and Associated Metabolic Disease (CIBERDEM), ISCIII, 28029 Madrid, Spain; 3Institute of Food Science, Technology and Nutrition (ICTAN-CSIC), 28040 Madrid, Spain; 4Pere Virgili Institute for Health Research (IISPV), 43007 Tarragona, Spain; 5Biosfer Teslab, 43201 Reus, Spain; 6Department of Electronic Engineering, Rovira i Virgili University (URV), 43003 Tarragona, Spain; 7National Centre for Biotechnology (CNB-CSIC), 28049 Madrid, Spain

**Keywords:** polyphenols, urine metabolites, type 2 diabetes, branched-chain aminoacids, untargeted metabolomics

## Abstract

Cocoa constitutes one of the richest sources of dietary flavonoids with demonstrated anti-diabetic potential. However, the metabolic impact of cocoa intake in a diabetic context remains unexplored. In this study, metabolomics tools have been used to investigate the potential metabolic changes induced by cocoa in type 2 diabetes (T2D). To this end, male Zucker diabetic fatty rats were fed on standard (ZDF) or 10% cocoa-rich diet (ZDF-C) from week 10 to 20 of life. Cocoa supplementation clearly decreased serum glucose levels, improved glucose metabolism and produced significant changes in the urine metabolome of ZDF animals. Fourteen differential urinary metabolites were identified, with eight of them significantly modified by cocoa. An analysis of pathways revealed that butanoate metabolism and the synthesis and degradation of branched-chain amino acids and ketone bodies are involved in the beneficial impact of cocoa on diabetes. Moreover, correlation analysis indicated major associations between some of these urine metabolites (mainly valine, leucine, and isoleucine) and body weight, glycemia, insulin sensitivity, and glycated hemoglobin levels. Overall, this untargeted metabolomics approach provides a clear metabolic fingerprint associated to chronic cocoa intake that can be used as a marker for the improvement of glucose homeostasis in a diabetic context.

## 1. Introduction

Type 2 Diabetes (T2D) is the most prevalent metabolic disease in the world, with a large socio-sanitary impact due to its chronic macro- and micro-vascular complications [1]. T2D is characterized by the elevation of blood glucose resulting from defects in insulin secretion, insulin action, or both. [2]. There are several glucose-lowering drugs available for the treatment of diabetes; however, most of them have potential adverse effects and are not completely effective in preventing the progression of the disease [3]. Therefore, more research in this area is crucial to develop alternative therapeutic agents or strategies that might reduce the harmful effect of diabetes and its associated complications.

Natural products rich in phenolic compounds represent promising candidates for the treatment of diabetes, due to their good effectiveness in glucose metabolism, and their low cost and toxicity [4]. Particularly, flavonoids, the most abundant phenolic compounds in fruits and vegetables, have been described as multitarget agents with great therapeutic potential for diabetes management [5]. In this regard, cocoa constitutes one of the richest sources of dietary flavonoids, mainly flavanols which is gaining importance in research since it has been proven to reduce the pathogenesis of diabetes and its complications [6]. Notably, our previous studies have demonstrated that cocoa supplementation improves the glucose metabolism in diabetic animals [7] and has beneficial effects on associated diabetic complications, mainly arterial stiffness [8] and nephropathy [9]. Several mechanisms have been described as being involved in these positive outcomes [10], including a possible prebiotic effect of cocoa [11]. However, the precise underlying mechanisms of these antidiabetic actions are not fully understood. In this sense, knowing how cocoa modulates metabolic pathways significantly affected in diabetes could be essential to further clarify its final biological actions.

Over the last years, metabolomics approaches have been successfully applied to evaluate metabolite alterations related to metabolic disease, such as diabetes [12], as well as to investigate the metabolic response to a nutritional intervention by characterizing the disturbed metabolites between control and treated groups [13]. Untargeted metabolomic approaches are especially useful, as they offer information on the levels of endogenous small molecule metabolites, presented in a biological sample without a priori information [14]. This comprehensive analysis unravels metabolite profile modifications and detects alterations in biological pathways and biochemical processes, providing further insight into the molecular mechanisms involved in the nutritional intervention. Particularly, nuclear magnetic resonance (NMR) is the most common analytical technique used to reveal profiles of metabolites involved in the tricarboxylic acid (TCA) cycle, amino acid, and carbohydrates metabolism, all of them suggested as significant pathways disturbed in diabetic models [15]. Indeed, the NMR study of biofluids has been proven to be useful to investigate the anti-diabetic activity of several plants’ bioactive compounds [15,16,17,18] as well as dietary flavonoids [19,20]. However, at present, there are no studies evaluating the metabolic impact of cocoa consumption in a diabetic setting, which could certainly help to better understand the mechanisms underlying its anti-diabetic activity.

Therefore, this study was aimed to identify potential metabolic changes that could be involved in the beneficial effect of cocoa on diabetes. To this end, we have performed an untargeted NMR-based metabolomics analysis along with a multivariate analysis in an in vivo model of T2D, the Zucker diabetic fatty (ZDF) rats. Our strategic approach has revealed significant differences in the urine metabolic profiles between control (ZDF) and cocoa supplemented (ZDF-C) diabetic animals. In addition, main differential metabolites and metabolic pathways affected by the cocoa intake in diabetic animals have been identified. Overall, these results provide new insights on the mechanism underlying the impact of cocoa on T2D, improving the functional understanding of its anti-diabetic effects.

## 2. Materials and Methods

### 2.1. Diets, Animals, and Experimental Design

Diets were prepared from an AIN-93G formulation (Panlab S.L., Barcelona, Spain). A cocoa rich-diet (10%) was produced by adding 100 g/kg of natural Forastero cocoa powder (a kind gift from Idilia Foods, Barcelona, Spain) to AIN-93G diet. The total polyphenolic content of the cocoa powder, as determined with the Folin-Ciocalteu method, was 2.4 g/100 g on a dry matter basis. The main cocoa flavonoids were determined by LC-MS, as previously described [8]. Monomeric epicatechin (382.0 mg/100 g) and catechin (115.2 mg/100 g) were the major flavanols in the extract, together with appreciable amounts of procyanidins B1 (35.2 mg/100 g) and B2 (132.2 mg/100 g). The resulting cocoa diet was isoenergetic and its composition is given in Appendix A.

Male ZDF rats and their Zucker lean controls (ZL) were purchased from Charles River Laboratories (L’arbresle, France) at 9 weeks of age. Animals were acclimated for one week under standard controlled conditions (21 °C ± 1 °C; 12 h day/night cycle). After that, the ZDF rats were randomly sorted into two groups (eight animals per group) that received the standard AIN-93G diet (ZDF), or the same control diet supplemented with 10% of cocoa (ZDF-C), for 10 weeks. The lean Zucker rats (ZL) (*n* = 6) received the standard AIN-93G diet. During the experiment, food and water were available *ad libitum*. Body weight and blood glucose in overnight fasted animals were followed weekly during the entire study and food intake was monitored three times per week. The animals were treated according to the European (2010/63/EU) and Spanish (RD 53/2013) legislation on Care and Use of Experimental Animals and the experiments were approved by the Ethics Committee from Comunidad de Madrid (PROEX 304/15).

### 2.2. Biochemical Determinations

At 20 weeks of age, the animals were fasted overnight, and blood samples were collected for biochemical analysis. Blood glucose was determined using an Accounted Glucose Analyser (LifeScan España, Madrid, Spain). Serum insulin and Hb1Ac were analyzed with ELISA kits (Rat Insulin, Mercodia, Uppsala, Sweden; HbA1c Kit Spinreact, BioAnalitica, Madrid, Spain). The fasting plasma concentrations of both glucose and insulin were used to calculate the indices of homeostatic model assessment of the insulin resistance (HOMA-IR) and insulin secretion (HOMA-B) according to the following formulas: HOMA-IR = fasting insulin (mU/mL) X fasting glucose (mM)/22.5 and HOMA-B = 20 X fasting insulin (mU/mL)/[fasting glucose (mM) − 3.5], respectively. Triacylglycerols (TG), HDL-Cholesterol and LDL-Cholesterol were determined in serum by kits (BioSystems, Madrid, Spain) as described elsewhere [8].

One week before the end of the study, a glucose tolerance test was performed. Briefly, after overnight fasting, 35% glucose solution (2 g/kg of body weight) (Sigma Chemical, Madrid, Spain) was administrated to the rats by intraperitoneal injection. Blood samples were obtained from the tail vein before the glucose load (t = 0) and at, 30, 60, 90, and 120 min, after the glucose administration and glucose levels were measured using an Accounted Glucose Analyzer (LifeScan España, Madrid, Spain). The integrated glucose response (area under the curve, AUC) over a period of 120 min after glucose overload was also calculated.

### 2.3. Preparation of Urine Samples and NMR Analysis

The 23 h urine samples were collected by means of metabolic cages at the end of the experiment, and were prepared and analyzed using standardized and optimized protocols, as previously described [21]. Briefly, the urine samples were thawed at room temperature before 540 μL of urine was mixed with 60 μL of buffer (KH_2_PO_4_, 1.5 M, pH 7.4 made up in ^2^H_2_O) containing 5.8 mM mM trimethylsilyl propionate (TSP) and 2 mM NaN_3_. Following centrifugation (12,000× *g*, 4 °C, 5 min) to remove solids, 550 μL of sample were transferred into 5 mm SampleJet NMR tubes and immediately loaded onto a refrigerated SampleJet robot (Bruker Biospin, Germany) and maintained at 4 °C until NMR analysis

The NMR area associated with the concentration of each metabolite was obtained after the spectral analysis by using an in-house lineshape fitting based on an algorithm developed to deconvolute the pre-processed NMR spectra by using Lorentzian and Gaussian functions to minimize the fitting error. The NMR areas were transformed into concentration units by using specific conversion factors depending on the proton numbers of the molecular structure generating the signal and a TSP internal standard as previously described [22]. Finally, the metabolite concentrations were normalized by using PQN normalization [23] to avoid urine dilution effects. Resonances assignment and metabolite identification was verified using the Chenomx and the Human Metabolome Database (HMDB).

### 2.4. Statistical and Multivariate Analysis

The data from biochemical parameters were tested for homogeneity of variances by the test of Levene; for multiple comparisons, one-way ANOVA was followed by a Tukey test when variances were homogeneous, or by the Tamhane test when variances were not homogeneous. All the data were presented as mean ± standard deviation (SD). For the metabolomics study, the median and interquartile range concentration of each metabolite was compared by using non-parametric Wilcoxon rank-sum text. A GraphPad Prism version 8.00 (GraphPad software, Inc., La Jolla, CA, USA) was used. The level of significance was *p* < 0.05.

Multivariate analysis was used to analyze the metabolomics differences of urine of the different groups. Principal components analysis (PCA) was performed to describe the metabolic profiles among groups. This unsupervised method allows us to identify the maximum number of uncorrelated principal components that together explain the maximum amount of variance in the NMR metabolomic data set. Prior to running the PCA, we auto scaled the input data and then transformed to create a composite score for each principal component. The loading analysis of the principal components was used to identify the most relevant molecular components. Orthogonal partial least squares discriminant analysis (OPLS-DA) was applied as the supervised regression modelling for discriminating groups. The validity of the models against overfitting was estimated by the parameter R2Y, and the predictive ability was described by Q2 values. Furthermore, the differential metabolites of variable importance in the projection (VIP) from the OPLS-DA were utilized to screen the biomarkers. Relationship strength between differential metabolites and physiological parameters was assessed using the two tailed Pearson’s correlation test. The correlation was considered significant when the absolute value of Pearson’s correlation coefficient r was > 0.5. Finally, identified metabolites were submitted to the metabolite set enrichment analysis (MSEA) module in MetaboAnalyst 5.0 using HMDB identifiers. The Kyoto Encyclopedia of Genes and Genomes (KEGG) was used as input.

## 3. Results

### 3.1. Effect of Cocoa Supplementation on Physiological Parameters

The metabolic characteristics of the ZL and ZDF animals are shown in Table 1. At 20 weeks of life, the body weight of ZDF and ZDF-C rats were significantly increased as compared to non-diabetic ZL animals. Likewise, both ZDF groups showed hyperglycemia, hyperinsulinemia, increased HbA1c levels, and glucose intolerance (AUC), which confirmed their hyperphagic and diabetic state. However, all these parameters were significantly improved in diabetic animals that were supplemented with cocoa. Moreover, cocoa intake was also effective in reducing insulin resistance (HOMA-IR) and glucose intolerance (AUC) in ZDF rats, as well as increasing beta-cell function (HOMA-B). On the contrary, the serum levels of HDL-Cholesterol, LDL-Cholesterol, and TG, were significantly elevated in both ZDF groups in comparison to the ZL group, and there were no differences between those animals fed with standard (ZDF) or with cocoa diet (ZDF-C). These results indicate that a cocoa rich diet significantly improved glucose homeostasis but not lipid profile, in 20-weeks old ZDF rats.

### 3.2. Metabolic Analysis of Urine Samples

An untargeted approach using ^1^H-NMR techniques was used to explore the metabolomics changes in the urine of ZL, ZDF and ZDF-C animals. A total of 37 metabolites were identified and quantified in the urine of lean and diabetic rats (Appendix A). A representative ^1^H-NMR spectrum of one of the urine samples with the metabolite assignment is shown in Appendix A.

To obtain a preliminary understanding of the overall differences in metabolites between groups, and the degree of variability among the samples within the group, we achieved an exploratory PCA on the entire dataset. As shown in Figure 1A, the first two components (PC1 and PC2) were enough to identify specific group molecular characteristics, explaining 55% of the variance in the data (37% and 18%, respectively). PC1 showed a clear separation among non-diabetic ZL and the two diabetic groups, driven by higher concentration of creatinine, 2-oxoisocaproate and nicotinamide-N-oxide in the ZL group, together with a lower concentration of glucose and 4-hydroxy phenylacetate (Figure 1B). On the other side, PC2 was not enough to separate ZDF-C and ZDF groups, but it can be achieved using PC1 and PC2 together. Altogether, our results strongly indicated that cocoa supplementation modify the urinary metabolic profile of ZDF diabetic animals.

### 3.3. Identification of Potential Biomarkers Associated with the Cocoa Ingestion in ZDF Rats

Based on this initial observation, the OPLS-DA supervised method was employed to explain the differences caused by cocoa intake in diabetic Zucker rats. As shown in Figure 2A, a clear separation between the two groups (ZDF and ZDF-C) was observed as demonstrated by the OPLS-DA scores scatter plot. Moreover, R2Y and Q2 parameters were relatively high with good predictive ability and reliability (R2Y = 0.812 and Q2 = 0.729, respectively), which was able to manifest the changed trend in metabolites between groups. In order to discover the potential metabolites contributing to the group classification, the variable importance in the projection (VIP) values of the obtained multivariate analysis OPLS-DA model was used. Establishing a cut-off VIP values above 1.0 and the significance level *p* < 0.05, we found 14 endogenous metabolites that could be identified as potential biomarkers (Figure 2B). For the cocoa intake, the most common metabolites enrichment were valine, isoleucine, leucine, 2-oxoglutarate, alanine, and hippurate. On the other hand, 3-indolelsulfate, D_unknow4 metabolite, acetoacetate, urea, 4-hidroxyphenillactate, suberate, formate, and glucose, were enriched in the diabetic control rats.

For these 14 metabolites, we further assessed their differential urine concentration between groups, yielding eight metabolites (valine, leucine, isoleucine, acetoacetate, urea, hippurate, 3- indolelsulfate, and D_unknow4) that were significantly modified by the cocoa intake (Figure 3).

### 3.4. Correlation Analysis between Urine Metabolites and Biomarkers Associated with Cocoa Intake

Next, we investigate the relationship between the significant metabolites identified in the urine of diabetic animals and the clinical parameters related to glucose and lipid metabolisms. To this end, a Pearson’s correlation analysis was conducted. The results were presented as a correlation heatmap, where green color indicates a positive correlation and red color indicates a negative one (Figure 4). A significant positive association was found between the levels of acetoacetate, 3-indolelsulfate, and the D_unknown4 metabolite and increased body weight, glycaemia, and HbA1c, as well as with insulin resistance (HOMA-IR) and glucose intolerance (AUC). However, valine and isoleucine were negatively correlated with all these parameters. Likewise, the levels of leucine showed significantly negative correlations with glucose, HbA1c, and AUC. The results also showed that urea exhibited a positive and significant association with the increased levels of glucose and HbA1c. Interestingly, hippurate levels, despite of all the other metabolites, were positively linked with an increase in the beta cell function (HOMA-B). On the other hand, increased LDL levels were positively associated with valine, isoleucine, and leucine.

### 3.5. Analysis of Biomarker Networks and Reconstruction of Metabolic Pathways

To further explore the metabolic pathways associated to the cocoa intake in diabetic animals, we performed a metabolite set enrichment analysis (MSEA) using the KEGG database as a framework (MetaboAnalyst 5.0). The enriched analysis showed the main metabolic routes modulated by the cocoa intake under diabetic conditions (Figure 5). Among them, butanoate metabolism, valine, leucine, and isoleucine degradation and biosynthesis, pantothenate and CoA biosynthesis, and synthesis and degradation of ketone bodies, were significantly enriched (*p* < 0.05) (Table 2). Therefore, these pathways represent the potential targeted pathways of cocoa intake in the diabetic condition. Based on the above results, a correlation network graph responding to the cocoa intake was constructed (Figure 6).

## 4. Discussion

The anti-diabetic potential of cocoa has previously been described both in pre-diabetic [7] and in diabetic ZDF animals [8]. Cocoa exerts these beneficial effects via multiple mechanisms, including antioxidant and anti-inflammatory effects, as well as by increasing both insulin secretion and insulin action [10]. Likewise, cocoa intake can modify the composition of the gut microbiota in ZDF rats, and these changes have been closely associated with the improved glucose homeostasis [11]. Herein, we show for the first time that chronic cocoa supplementation for 10 weeks significantly modified the urinary profile of diabetic fatty rats. These findings are in concordance with those of Massot-Cladera et al. [24] who exposed that the consumption of a cocoa-rich diet for 3 weeks resulted in a different urinary metabolic pattern in normal Wistar rats. In contrast, in a recent human intervention study, chronic consumption ofprocyanidin-rich cocoa (10 weeks) was found to have a marginal impact on serum metabolome in male endurance athletes [25]. Likewise, the biological response of a free-living population to the daily consumption of dark chocolate for two weeks only had a significant metabolic impact in subjects with essential high anxiety [26]. Together, these data seem to indicate that the effects of cocoa on the human metabolome are highly dependent on the stress situations of individuals, highlighting the beneficial implications of cocoa on the metabolic response to stress. According to this, our results further support the positive effects of the cocoa consumption on stress-associated metabolic abnormalities, like those that appear in animals with diabetes.

In the present study, we applied NMR-based metabolomics and multivariate analysis, including PCA and OPLS-DA, to study the metabolic profile of ZDF rats and discriminate the protective effect of a cocoa-rich diet. A total of 14 potential biomarkers associated with diabetes and obesity were identified, which were related to disturbed metabolic pathways involving valine, leucine and isoleucine biosynthesis, synthesis and degradation of ketone bodies, butanoate metabolism, pantothenate and CoA biosynthesis and aminoacyl-tRNA biosynthesis, among others. The chronic supplementation of diet with 10% of cocoa powder interfered in these metabolic pathways by restoring potential biomarkers back to normality or even improving them, which might be part of the protective mechanism of cocoa on the disrupted diabetic metabolism.

Overt diabetes is characterized by an increased conversion of proteins into glucose via hepatic gluconeogenesis, contributing to worsen the hyperglycemia. In agreement, many reports have shown a positive correlation between insulin resistance and elevated plasma levels of branched chain amino acids (BCAAs, such as valine, leucine, isoleucine) [27,28,29]. The mechanistic explanation for this correlation seems to include an enhanced turnover of BCAAs, rather than an increase in their biosynthesis [28]. Moreover, blood accumulation of BCAAs leads by itself to the development of further insulin resistance [27,30], whereas the decreased dietary consumption of BCAAs without calorie restriction enhances energy expenditure and improves insulin sensitivity in mice [31].

However, not many studies have examined the level of amino acids in the urine of diabetic patients. Salek et al. [32] showed in db/db diabetic mice a decrease in the excretion of several amino acids, including valine and leucine. Interestingly, an increase of the amino acid concentration in the urine has been reported in diabetic patients following rosiglitazone treatment [33], suggesting that this change could be associated to the pharmacological treatment that, by reducing insulin resistance, also decreases the demand for amino acids as substrates for gluconeogenesis. Likewise, urinary metabolomic data indicate that physical activity improves insulin sensitivity in diabetic patients and raises BCAA excretion [34]. In the present study, we have observed an inverse correlation between the urine concentration of BCAAs, mainly valine and isoleucine, with body weight, HOMA-IR and glucose tolerance test AUC. Therefore, the increased amino acid excretion observed in cocoa-fed diabetic rats is suggestive of a less active hepatic gluconeogenesis and, accordingly, improvedperipheral sensitivity to insulin [35].

Consistent with the common increased protein breakdown in diabetic patients [32,36], the high levels of urea, the final product of amino acid catabolism, were also present in ZDF urinary samples in comparison with non-diabetic rats, but cocoa administration entirely normalized this parameter. The mechanism involved in this phenomenon might include the amelioration of insulin resistance and the improvement of insulin production and secretion, as we have previously described [7].

In addition, leucine and isoleucine are also considered ketogenic amino acids since they are degraded entirely or in part into acetoacetyl-CoA and/or acetyl-CoA. In the liver, acetoacetyl-CoA is converted to acetoacetate and then to acetone and β-hydroxybutyrate. Their ability to form ketone bodies is particularly evident in inadequately controlled diabetes, in which the liver produces large amounts of ketone bodies from both fatty acids and the ketogenic amino acids. In contrast, by observing the reduction of acetoacetate in the urine of ZDF-C rats, we can speculate that the cocoa rich-diet is able to prevent the enhancement of ketone body synthesis associated to insulin resistant states. Moreover, the low levels of urinary acetoacetate in the present study are correlated with the low levels of T2D biomarkers, such as, body weight, glycemia, insulinemia, HbA1c, HOMA-IR values, or glucose intolerance.

Interestingly, metabolite enrichment analyses and KEGG pathway map also revealed significant differing metabolites belonging to the butanoate metabolism, including not only acetoacetate but also 2-oxoglutarate. Butanoate metabolism, also known as butyrate metabolism, outlines the metabolic destiny of short-chain fatty acids or short-chain alcohols produced by the intestinal bacterial fermentation of fiber. Many of these molecules are ultimately used in the production of ketone bodies, as the aforementioned acetoacetate, the synthesis of lipids, or as precursors of the tricarboxylic acid (TCA) cycle or glutamate synthesis. The supplementation of diet for 10-weeks with cocoa significantly increased urine 2-oxoglutarate levels in ZDF rats compared with the values observed in ZL animals (*p* = 0.001), but not in ZDF rats receiving the standard diet. Thus, the increased presence of 2-oxoglutarate, one of the most important TCA cycle intermediates, may be due to an enhancement in overall mitochondrial biogenesis induced by the cocoa, or increased TCA substrate availability related to the previously mentioned amelioration of ketogenesis and gluconeogenesis. The effects of cocoa and its constituents on energy metabolism have been extensively studied [24,26] with dissimilar results. Massot-Cladera et al. [24] observed that the administration of a diet containing 10% cocoa to healthy Wistar rats decreased urinary excretion of energy intermediates, such as citrate, 2-oxoglutarate, and N-methylnicotinamide, which is opposite to our current data obtained in diabetic rats. These differential results highlight that the effects of cocoa intake on health are highly determined by the (patho)physiological state of individuals.

Moreover, diabetic rats receiving the cocoa-rich diet also excreted greater amounts of the glycine conjugate of benzoic acid, hippurate, compared with those receiving the standard diet. Although hippurate is a normal constituent of the endogenous urinary metabolite profile, enhanced excretion has been associated with gut microbial metabolism of polyphenol-rich components of the diet, such as vegetables, tea, coffee [37,38], and cocoa [24]. It has been proposed that, in addition to the availability of precursors in the diet, the absence or presence of urinary hippurate is also influenced by variations of the intestinal microbiota. In agreement, we and others have previously described a marked gut dysbiosis in ZDF rats [11,39], whereas chronic cocoa supplementation modified the gut microbiota to a healthier profile in diabetic rats [11]. Interestingly, decreased hippurate excretion was found in a ^1^H-NMR-based metabonomic analysis of urine samples from obese individuals [40] or type 2 diabetic patients [32], as well as in Zucker (fa/fa) obese rats compared to Zucker (fa/-) lean controls [32]. On the contrary, weight loss has been linked to higher hippurate excretion in both humans [40] and animals [41]. Consistent with this, we have described that a cocoa-rich diet is able to reduce the body weight of ZDF rats without modifying the daily food intake, probably by variations in the activity of gut microbiota and basal energy expenditure [11]. The beneficial effects of increased hippurate production on glucose metabolism needs further studies, but it has been reported that its precursor, hippuric acid, potentiates glucose stimulated insulin secretion from beta cells [42], which is in agreement with the positive correlation observed herein between hippurate and HOMA-B values.

Conversely, cocoa-fed diabetic rats excreted lower amounts of other metabolites arising from gut microbial-host metabolism, such as 3-indolesulfate, a derived product of tryptophan [43]. The study of Dou et al. [44] showed that the uremic solute 3-indolesulfate was able to cause oxidative stress in endothelial cells by increasing nicotinamide adenine dinucleotide phosphate (NADPH)-oxidase activity and ROS production. Obesity and diabetes are considered chronic inflammatory diseases closely related with oxidative stress [45], and cocoa consumption has long been demonstrated to have antioxidant effects and to improve the metabolic profile in diabetes [6,8]. Mechanistically, we previously reported that cocoa intake prevented ROS production through downregulating the protein expression of NADPH oxidase in ZDF rat aorta, which may be beneficial in decreasing the risk of the cardiovascular disease associated to diabetes [8]. Thus, our present results, in agreement with those of other authors [24], indicate that the reduced urinary concentration of 3-indolesulfate observed in ZDF-C rats reflects the antioxidant capacity of cocoa.

## 5. Conclusions

This study shows, for the first time, that a controlled and realistic supplementation of diet with cocoa powder induced changes in the urinary metabolome of diabetic animals (Figure 6). The use of a holistic approach, such as untargeted metabolomics, allowed us to disclose how cocoa can modulate the endogenous metabolism, thus providing further insights into the positive effects that chronic cocoa intake has in a diabetic context.

Specifically, the novelty of this study is that the treatment of diabetic animals with cocoa markedly increased urinary BCAAs levels, while reducing acetoacetate values, possibly indicating a decrease in gluconeogenesis as well as in ketogenesis and, therefore, an improvement of insulin sensitivity and glycemic control (Figure 6). It is conceivable that improved insulin action would also lead to the stimulation of protein synthesis and the further reduction of circulating BCAAs. Since BCAA catabolic intermediates feed into other metabolic pathways, such as the TCA cycle, when in excess, BCAAs may disturb the mitochondrial function. Consequently, it is reasonable to infer that cocoa’s beneficial effects could be due in part to its impact on energy metabolism.

Finally, it is worth noting that while the ZDF rats have an obesity phenotype, not all changes observed in this model may be generalizable to the entire human forms of T2D. Additional studies are needed to determine which of these pathways and changes observed in this model, in addition to an elevation of BCAA excretion, are also observed in humans.

## Figures and Tables

**Figure 1 nutrients-14-04127-f001:**
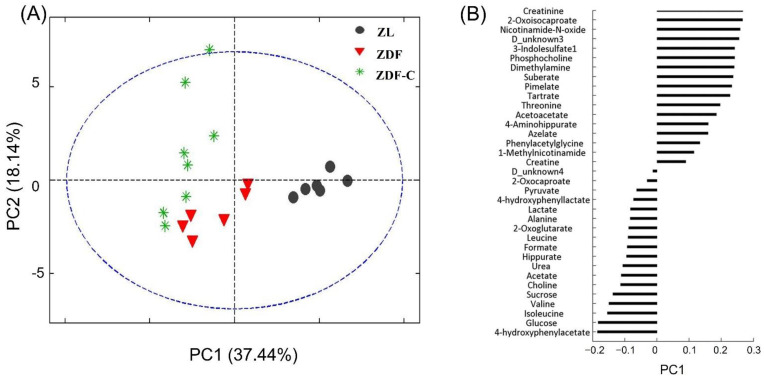
(**A**) Principal components analysis (PCA) score plots from urine samples from normal lean (ZL), diabetic (ZDF), and cocoa-supplemented diabetic (ZDF-C) rats, at 20 weeks of life. (**B**) Loading plot of PC1.

**Figure 2 nutrients-14-04127-f002:**
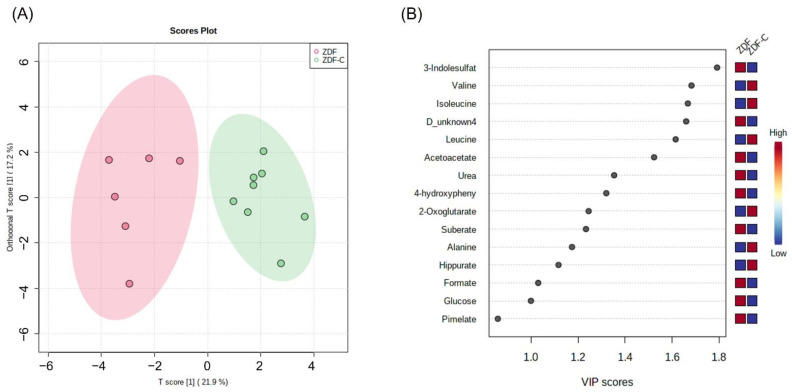
(**A**) Orthogonal partial least squares discriminant analysis (OPLS-DA) score plots from urine samples of diabetic (ZDF) and cocoa-supplemented diabetic (ZDF-C) rats at 20 weeks of life. (**B**) VIP values derived from OPLS-DA. The blue and red boxes on the right indicate whether the mean metabolite abundance is increased (red) or decreased (blue) in ZDF-C vs. ZDF.

**Figure 3 nutrients-14-04127-f003:**
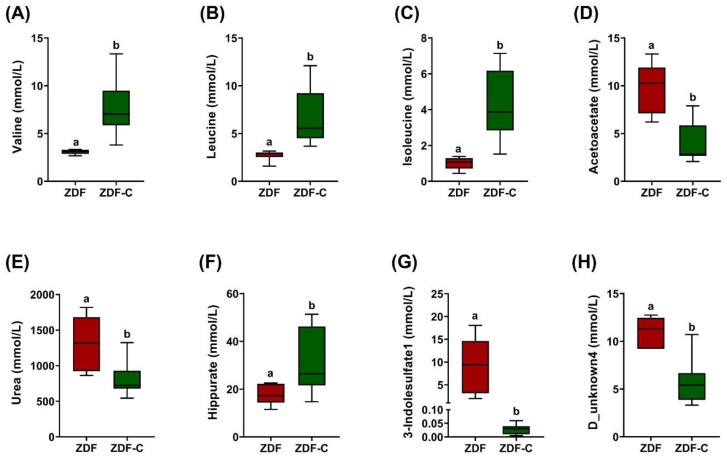
Altered urine metabolites between the diabetic (ZDF) and cocoa-supplemented diabetic (ZDF-C) rats at 20 weeks of life: (**A**) valine, (**B**) leucine, (**C**) isoleucine, (**D**) acetoacetate, (**E**) urea, (**F**) hippurate, (**G**) 3- indolelsulfate, and (**H**) D_unknow4 metabolite levels. Data represent the means ± SD of 6–8 animals. Different letters denote statistically significant differences, *p* < 0.05.

**Figure 4 nutrients-14-04127-f004:**
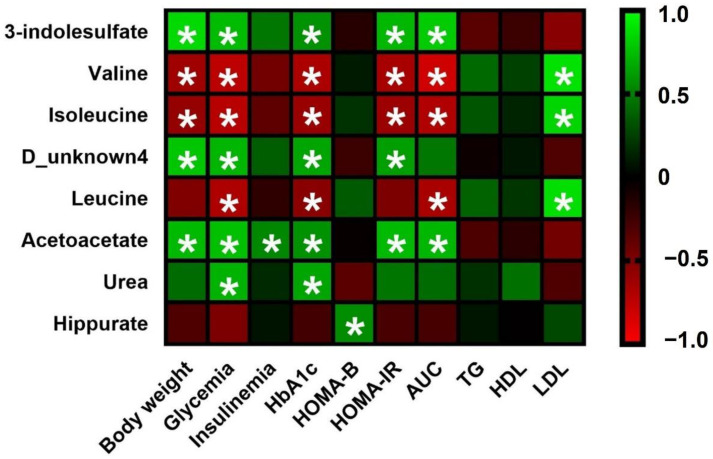
Heatmap of correlation between the main significantly altered urine metabolites and biochemical biomarkers related to diabetes. Pearson correlation values were used for the matrix. Green color indicates a positive correlation, whereas red color indicates a negative correlation; the intensity of the color represents the degree of association. * Represents adjusted *p* < 0.05.

**Figure 5 nutrients-14-04127-f005:**
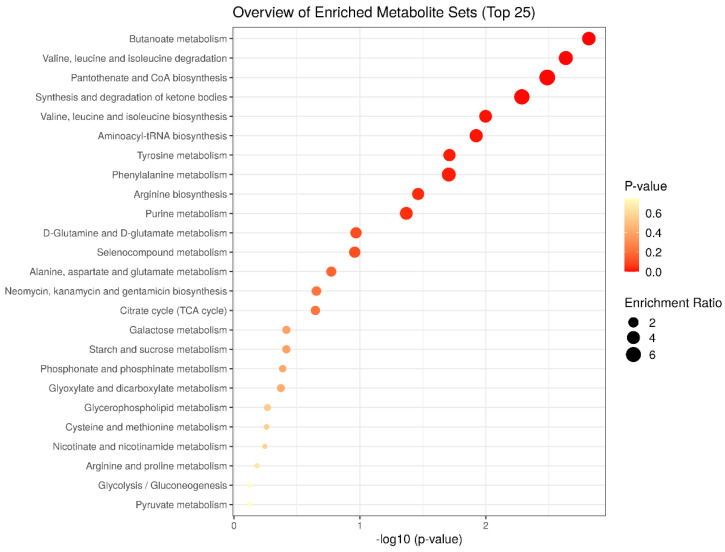
Metabolite set enrichment analysis of significantly altered urine metabolites according to the KEGG database.

**Figure 6 nutrients-14-04127-f006:**
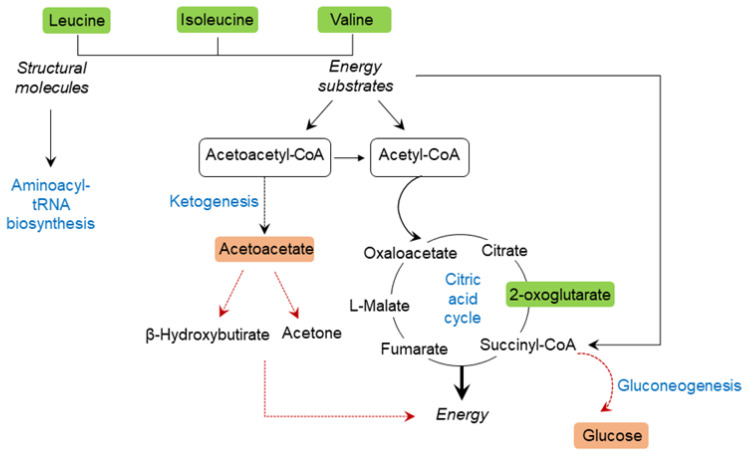
Schematic representation depicting the interrelationships of the disturbed metabolic pathways identified by ^1^H NMR urine analysis. Green (significantly increased), and red (not significantly increased), as compared to ZDF (*p* < 0.05 is significant).

**Table 1 nutrients-14-04127-t001:** Biological parameters of Zucker lean rats (ZL), Zucker diabetic fatty rats fed with control diet for 10 weeks (ZDF), and Zucker diabetic fatty rats fed with cocoa diet for 10 weeks (ZDF-C). Data represent the means ± SD of 6–8 animals. Different letters denote statistically significant differences, *p* < 0.05.

	ZL	ZDF	ZDF-C
Body weight (g)	329 ± 41 ^a^	444 ± 82 ^b^	400 ± 12 ^c^
Glucose levels (mg/dL)	86 ± 90 ^a^	238 ± 71 ^b^	118 ± 11 ^c^
Insulin levels (ng/mL)	0.40 ± 0.02 ^a^	4.36 ± 0.50 ^b^	1.14 ± 0.22 ^c^
HbA1c (%)	4.38 ± 0.19 ^a^	10.40 ± 1.58 ^b^	6.09 ± 0.79 ^c^
AUC (mmol/L/min)	1803 ± 96 ^a^	4044 ± 60 ^b^	3049 ± 33 ^c^
HOMA IR	2.60 ± 0.20 ^a^	90.96 ± 14.26 ^b^	12.22 ± 1.36 ^c^
HOMA B	149.81 ± 22.21 ^a^	169.52 ± 46.20 ^a^	227.83 ± 14.32 ^b^
TG (mmol/L)	0.39 ± 0.09 ^a^	2.74 ± 0.24 ^b^	2.87 ± 0.31 ^b^
HDL-Cholesterol (mmol/L)	2.27 ± 0.22 ^a^	2.85 ± 0.37 ^b^	3.08 ± 0.32 ^b^
LDL-Cholesterol (mmol/L)	0.92 ± 0.09 ^a^	2.28 ± 0.33 ^b^	2.82 ± 0.31 ^c^

**Table 2 nutrients-14-04127-t002:** Top pathways enriched with metabolites having significantly altered abundance in Zucker diabetic fatty rats fed with cocoa diet (ZDF-C), as identified by the pathway analysis using MetaboAnalyst.

Top Pathways	Total Compounds	Hits	*p* Value	FDR *	Metabolites Identified
Butanoate metabolism	15	2	0.0015	0.026	Acetoacetate, 2-oxoglutarate
Valine, leucine and isoleucine degradation	40	5	0.0023	0.026	Isoleucine, Leucine, Valine, Acetoacetate, 4-Methyl-2-oxopentanoate
Pantothenate and CoA biosynthesis	19	1	0.0032	0.026	Valine
Synthesis and degradation of ketone bodies	5	1	0.0051	0.031	Acetoacetate
Valine, leucine and isoleucine biosynthesis	8	5	0.01	0.047	Isoleucine, Leucine, Valine, Threonine, Acetoacetate,
Aminoacyl-tRNA biosynthesis	48	5	0.019	0.047	Isoleucine, Leucine, Valine, Alanine, Lysine

* FDR is the *p* value adjusted using False Discovery Rate.

## Data Availability

Data are available upon request to the authors.

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
