# Peer review of "Urinary Metabolomics Study on the Protective Role of Cocoa in Zucker Diabetic Rats via 1H-NMR-Based Approach"

_nutrients, 2022, doi:10.3390/nu14194127_

Round 1

Reviewer 1 Report

The authors performed untargeted NMR-based urine metabolomics analysis and multivariate analysis to identify potential metabolic changes involved in the beneficial effect of cocoa on diabetes using the Zucker diabetic fatty (ZDF) rats. Cocoa supplementation produced significant changes in the urine metabolome, decreased serum glucose levels, and improved glucose metabolism in ZDF rats. In addition, a combination of metabolomic and pathway analyses suggested that cocoa decreases gluconeogenesis and ketogenesis and activates mitochondrial biogenesis. They concluded that their metabolomic approach provides further insight into the positive effects of diabetes.

The reviewer thinks that the manuscript is well-written. The findings are interesting for the reader of Nutrients. On the other hand, the reviewer could not find any supplemental materials, making it impossible to review strictly. In addition, the manuscript seems to appeal untargeted metabolomic approach, not provide a solid conclusion from the experimental results. Therefore, the reviewer asks to modify the abstract and conclusion section so that the reading audience could get the authors’ novel findings using untargeted metabolome research from the view of the beneficial effect of cocoa against diabetes.

Author Response

The authors performed untargeted NMR-based urine metabolomics analysis and multivariate analysis to identify potential metabolic changes involved in the beneficial effect of cocoa on diabetes using the Zucker diabetic fatty (ZDF) rats. Cocoa supplementation produced significant changes in the urine metabolome, decreased serum glucose levels, and improved glucose metabolism in ZDF rats. In addition, a combination of metabolomic and pathway analyses suggested that cocoa decreases gluconeogenesis and ketogenesis and activates mitochondrial biogenesis. They concluded that their metabolomic approach provides further insight into the positive effects of diabetes.

The reviewer thinks that the manuscript is well-written. The findings are interesting for the reader of Nutrients.

We would like to thank the Reviewer for the careful reading of the manuscript, as well as the encouraging comments and helpful suggestions.

1) On the other hand, the reviewer could not find any supplemental materials, making it impossible to review strictly.

We would like to apologize with the Reviewer since it has been a mistake by our part due to we did not upload the ‘Supplementary file’ when we submitted the first version of the manuscript. This file has been now included in the reviewed version.

2) In addition, the manuscript seems to appeal untargeted metabolomic approach, not provide a solid conclusion from the experimental results. Therefore, the reviewer asks to modify the abstract and conclusion section so that the reading audience could get the authors’ novel findings using untargeted metabolome research from the view of the beneficial effect of cocoa against diabetes.

We are not really sure about Reviewer’s indication regarding robustness of the results obtained with untargeted metabolomics approaches. Indeed, we consider that untargeted metabolomics is a very useful tool that offers very reliable information of the levels of endogenous small molecule metabolites present in a biological sample without a priori information. In this way we have expressed our opinion in the Introduction section (lines 68-73). Notwithstanding, and following Reviewer’s suggestion we have modified the abstract (lines 33-37) and conclusion (lines 468-469) in order to highlight the novelty of the present study for the reader.

Reviewer 2 Report

It is better to include the composition of Cocoa in the main text and calculate the total and separate amount of polyphenols or procyanidins (C/EC/B2) in the powder. 

In table 1, glycemia and insulinemia here is not correct, they should be changed to glucose and insulin levels. Also, keep all values consistent with 2 decimals. The second column and the third row, there is a mistake. please change this table carefully.

In table 1, the legend should not write 20-weeks Zucker rats, authors need to mention how you it was supplemented with Cocoa.

line 61, what is Overts?

Line 95, Zucker diabetic fatty is ZDF, in line 98, what is ZDF diabetic rats? additionally, you have already abbreviated ZDF, ZDF-C in the introduction section, so in methods section, do not need to write the full name again. please check carefully.

Line 118, what is HDL-Chol and LDL-Chol? Also, if the term only appears once in the text, you do not need to abbreviate it, like the GTT, TSP, NADPH...

Author Response

We would like to want to thank the Reviewer for the careful reading of the manuscript, as well as their interesting comments and suggestions.

1) It is better to include the composition of Cocoa in the main text and calculate the total and separate amount of polyphenols or procyanidins (C/EC/B2) in the powder.

In agreement with the Referee, the composition of the cocoa powder has been included in material and methods in the revised manuscript (lines 97-102).

2) In table 1, glycemia and insulinemia here is not correct, they should be changed to glucose and insulin levels. Also, keep all values consistent with 2 decimals. The second column and the third row, there is a mistake. please change this table carefully.

Following the Referee advice, we have double-checked the table 1 and we have corrected all noted errors.

3) In table 1, the legend should not write 20-weeks Zucker rats, authors need to mention how you it was supplemented with Cocoa.

The legend of Table 1 has been corrected according to the indications of the Reviewer.

4) Line 61, what is Overts?

We would like to apologize, as it was a grammatical error. In the revised manuscript, it has been replaced by the correct word (over). 

5) Line 95, Zucker diabetic fatty is ZDF, in line 98, what is ZDF diabetic rats? additionally, you have already abbreviated ZDF, ZDF-C in the introduction section, so in methods section, do not need to write the full name again. please check carefully.

We totally agree with the Reviewer’s consideration, ZDF by itself implies that the animals are diabetics. Accordingly, we have revised the whole document to eliminate redundancies. Furthermore, we have removed the full name of ZDF and ZDF-C in the material and methods section.

6) Line 118, what is HDL-Chol and LDL-Chol? Also, if the term only appears once in the text, you do not need to abbreviate it, like the GTT, TSP, NADPH...

We would like to apologize again for the inappropriate abbreviation used to make reference to cholesterol. Thus, we have changed Chol by Cholesterol. Moreover, following Reviewer’s comments, we have double-checked all abbreviations to remove those that only appear once in the text. Nevertheless, we have kept the abbreviations TSP and NADPH since they appear twice in the manuscript.